# Memory T-Cells Contribute to Calcium Release from Bones during Lactation in Mice

**DOI:** 10.3390/nu16193289

**Published:** 2024-09-28

**Authors:** Di Wu, Anna Cline-Smith, Brady Chrisler, Brittani Lubeck, Ajit Perla, Sumona Banerjee, Ida Fan, Rajeev Aurora

**Affiliations:** Department of Molecular Microbiology and Immunology, Saint Louis University School of Medicine, 1100 S Grand Blvd, Saint Louis, MO 63104, USA; andy.wu@health.slu.edu (D.W.); anna.smith@health.slu.edu (A.C.-S.); brady.chrisler@health.slu.edu (B.C.); brittani.lubeck@health.slu.edu (B.L.); ajit.perla@health.slu.edu (A.P.); sumona.banerjee@health.slu.edu (S.B.); xf1@williams.edu (I.F.)

**Keywords:** TNFα, IL-17A, postpartum, bone resorption, IL-7, IL-15

## Abstract

**Objective:** Milk production during lactation places a high demand for calcium that is fulfilled both from maternal bone resorption and diet. While it is known that mammary gland-derived PTHrP drives bone resorption during lactation, the impact of postpartum estrogen loss on bone has been unclear. **Methods:** We used a case-control study design to test the effect of estrogen loss in lactating mice. Results: In the present study, we show for the first time that estrogen loss during lactation activates memory T-cells (T_M_) to produce TNFα and IL-17A to aid in bone resorption and calcium release. Our studies reveal a new mechanism for the release of calcium from bone postpartum. The findings provide several new insights. First, the immune system plays a critical role in milk production postpartum. Second, evolutionarily, the pathway serves the physiological purpose of increasing bone resorption to release calcium for breastmilk production postpartum but becomes maladaptive postmenopause, leading to osteoporosis. Finally, these results highlight the crosstalk between the brain–bone–breast–endocrine axis and the immune system during lactation.

## 1. Introduction

The rapid skeletal growth of the developing fetus and the newborn creates a high demand for calcium during pregnancy and lactation. Approximately 2–3% of a mother’s calcium is transferred to the fetus in the later stages of pregnancy, primarily in the second and third trimesters. During lactation, about 300–400 mg of calcium per day is allocated to breastmilk [1,2,3]. Due to this increased demand, the body activates several mechanisms such as enhancing renal calcium retention, boosting intestinal absorption, and increasing bone resorption to meet that demand [4]. Despite these physiological adaptations, there is typically a decrease in bone mineral density by about 3% during lactation, which is counterbalanced by elevated levels of dihydroxy-vitamin D; fluctuations in parathyroid hormone (PTH), growth hormone, prolactin, and estrogen; and changes in nutrition, body weight, and lifestyle. The current view holds that the regulation of mineral homeostasis during lactation is primarily influenced by PTH-related protein (PTHrP) levels and a hypoestrogenic state [5].

In brief, the suckling of the baby stimulates the release of oxytocin and prolactin from the pituitary gland and subsequently leads to the release of PTHrP from the mammary glands to promote bone resorption [6]. Both static and dynamic histomorphometry have demonstrated increased bone turnover during lactation in favor of resorption [7]. Both osteoclast-driven trabecular resorption and osteocytic osteolysis of cortical bone contribute to calcium release [8,9,10,11,12]. The calcium released from the bone enters circulation and is transferred into the breast milk by the mammary glands to support the growth of the newborn. After weaning, the maternal skeleton typically recovers to the pre-pregnancy state in terms of bone mass [13]. 

While the effect of the brain–breast–bone axis is well established, the effect of estrogen (E_2_) depletion postpartum has not been investigated in detail. E_2_ regulates cell activities by multiple mechanisms. E_2_ can regulate transcription via binding to its cognate receptors (ESR1 or ESR2) that are members of the nuclear hormone receptor superfamily [14]. At the organismal level, E_2_ is a key component of the reproductive system in women and thus regulates key systems, including the immune system. During pregnancy, as the fetus is allogenic, the maternal immune system must be regulated to prevent immunogenic responses to both the placenta and the fetus while not immunocompromising the mother. Therefore, pregnancy, postpartum, and menopause could be distinct physiological states compared to normal menstrual cycling. Here, we investigated whether this E_2_ loss-dependent T_M_ activation plays a role in osteolysis during lactation.

We previously identified a pathway of how E_2_ regulates the immune system. While a hyperestrogenic state promotes tolerance, the depletion of E_2_ by ovariectomy (OVX) leads to the production of TNFα and IL-17A by T_M_ to promote bone loss [15]. T_M_ rely on IL-7 and IL-15 for homeostasis. In the bone marrow, a subset of dendritic cells (DCs) produces IL-7 and IL-15 [15]. E_2_ induces the apoptosis of IL-7^+^DC by inducing FasL. With the decline of E_2_, FasL is no longer induced, leading to increased numbers of DCs and higher concentrations of IL-7 and IL-15. These two cytokines together, but not individually, promote TNFα and IL-17A production in a subset of T_M_, which subsequently activate bone erosion [15]. To validate the role of IL-15 in activating T_M_ post-OVX, we ablated the IL-15 receptor alpha chain in T-cells by crossing IL15RA^f/f^ mice to T-cell-specific CRE mice (Lck-Cre) to generate the IL15RA^ΔΤ^ strain. These mice developed normal levels of T_M_, as expected [15]. Importantly, E_2_ loss did not lead to TNFα and IL-17A induction in T_M_ in the IL15RA^ΔT^ mice [15]. Thus, IL15RA^ΔT^ mice allow us to distinguish between the cell autonomous effect of E_2_ loss and T-cell-mediated inflammation on bone cells. We used these IL15RA^ΔT^ mice in this study to validate the effect of T_M_ on bone resorption during lactation.

## 2. Materials and Methods

*Animal use statement:* All animals used in this study were maintained in the Department of Comparative Medicine at Saint Louis University School of Medicine in accordance with institutional and Public Health Service guidelines. All mice were housed in microisolator cages in specific pathogen-free (SPF) environments. Weanling mice were maintained on rodent chow 5LOB and breeder trios were fed on breeder diet 5058 (both diets are from Lab Diet, Richmond, IA, USA). The Saint Louis University School of Medicine Institutional Animal Care and Use Committee (IACUC) approved all procedures performed on the mice (protocol 2072).

*Mice:* C57BL/6J (model 000664) mice were purchased from The Jackson Laboratory (Bar Harbor, ME, USA) and bred in-house. Breeding trios of IL-7^CFP^ reporter [16] mice were generously provided by Dr. Scott Durum (NIH-NCI, Bethesda, MD, USA) and bred in-house. The IL15RA^ΔT^ mice were previously generated [15] by crossing IL15RA-floxed (model 022365) [17] and Lck-Cre Tg540-I (model 006889) [18] mice, and were maintained in-house.Breeder trios (two nulligravid females and a male) were set up at 8–10 weeks of age. For the postpartum experiments in this study, the male was removed from the cage when the females were visibly pregnant and prior to delivery. The females were allowed to nurse pups normally. For postpartum harvests, both the nursing dam and her litter were euthanized on d12 postpartum. Nursing females were given 0.05 mg over a 21-day release of 17β-estradiol pellets (E-121, Innovative Research of America, Sarasota, FL, USA) subcutaneously in the loose skin of the scruff neck between d0 and d2 postpartum. Next, 1.5% isoflurane was used to initiate anesthesia and for maintenance. The pellet was placed using a 10G precision trocar, and closed using a single skin staple (3M, Maplewood, MN, USA; cat# DS-15).

*Flow cytometry: *Antibodies used in this study are listed in Table 1. For staining, cells were resuspended with 50 μL of BD Horizon^™^ Brilliant Stain Buffer (cat# 566349, BD Biosciences, Franklin Lakes, NJ, USA) and then incubated for 30 min at room temperature with fluorophore-conjugated antibodies protected from light. For intracellular staining (ICS), cells were washed, fixed with 4% paraformaldehyde, permeabilized with 0.1% Triton-X100, and stained overnight at 4 °C. Cells were washed, fixed with 1% paraformaldehyde, and analyzed on a LSRII instrument with FACS Diva 9.0 (BD Biosciences) software. Gates were determined using a combination of single-color and fluor-minus-one controls. Data analyses were performed with FlowJo software (version 10.8.1; FlowJo, LLC, Ashland, OR, USA).

*Immunofluorescence:* Femurs and tibias were fixed in 4% paraformaldehyde (PFA) overnight (16–24 h). The bones were washed briefly with PBS and submitted to Washington University Musculoskeletal Histology and Morphometry Core for decalcification and sectioning. Tibias were decalcified for 10–14 days in EDTA embedded in paraffin and sectioned at 5 μm thickness. Immunofluorescence (IF) staining was performed according to established protocol [19]. Briefly, slides were deparaffinized with Xylene and rehydrated in decreasing concentrations of ethanol (100%, 95%, 70%, 50%, 30%, and DI-H_2_O) in Coplin jars. Antigen retrieval was performed at 55 °C in citrate buffer (cat# C9999, Sigma-Aldrich, Saint Louis, MO, USA), and Background-Sniper (cat# BS966, BioCare Medical, Pacheco, CA, USA) was used for blocking. The primary antibodies used were rabbit anti-mouse IL-7 (cat# ab9732, Abcam, Cambridge, MA, USA), rat anti-mouse IL-15 (cat# MAB477, eBioscience, Santa Clara, CA USA), and Armenian hamster anti-mouse CD11c (cat# ab33483, Abcam). The secondary antibodies used in this study were goat anti-rabbit-AF594 (cat# Ab150080, Abcam), donkey anti-Armenian hamster-AF488 (cat# Ab173003, Abcam), and goat anti-rat-AF647 (cat# A21247, Invitrogen, Waltham, MA, USA). The slides were imaged using Leica TCS SP8 confocal microscopy (Leica, Wetzlar, Germany) at 40× magnification. Image analyses were performed using FIJI/ImageJ2 v.2.14.02/1.54f (https://imagej.net/software/fiji/downloads (accessed on 7 July 2023)) by an operator blinded to the experiments.

*TUNEL assay:* Formalin-fixed, paraffin-embedded bone sections were TUNEL-stained per the manufacturer’ instructions (in situ cell death detection kit TMR red, cat# 12156792910, Roche Diagnostics, Mannheim, Germany). Quantitation was performed as described for IF.

*In vitro DC culture:* The spleen was harvested in cold PBS-1, injected with 100 μL of Liberase TL (cat# 05401020001, Roche Diagnostics) at 125 μg/mL, and incubated at 37 °C for 10 min. The spleen was chopped into smaller pieces and further incubated in 3 mL of PBS-1 (PBS + 1% FBS) containing Liberase TL at 125 μg/mL at 37 °C for 30 min, and vortexed for 30 sec every 5 min. The spleens were passed through a 70 μ cell strainer with the aid of a syringe plunger washed with 5 mL of cold PBS-1. RBCs were lysed using BD Pharm Lyse™ (cat# 555899, BD Biosciences), and remaining cells were cultured in DMEM containing 10 nM 17β-estradiol (cat# E8875, Sigma-Aldrich) at 37 °C for 4 h. Cells were then analyzed via FACS.

*In vitro T-cell culture:* T-cells were isolated from BMC using Anti-CD4(L3T4) and anti-CD8a(Ly-2) microbeads (cat# 130–117-043 and cat# 130-117-044, Miltenyi Biotec, Bergisch Gladbach, Germany). T-cells were cultured in complete T-cell growth media (RPMI supplemented with 10% heat-inactivated FBS, Penicillin, Streptomycin, 2 mM glutamine, 1 mM sodium pyruvate, non-essential amino acids, and 55 μM β-mercaptoethanol; all from Sigma-Aldrich, St. Louis, MO, USA) for 2 days prior to cytokine treatment.

*Serum marker assay:* Nursing dams were fasted with their offspring overnight prior to blood collection. Blood (50 μL) was obtained via the submandibular vein and allowed to clot for 1 hour at room temperature. Serum was collected by spinning down the cell pellet at 10,000 g for 10 min, flash freezing, and storing at −80 °C. Serum C-terminal telopeptide of type 1 collagen (serum CTX) was measured using a competitive ELISA according to the manufacturer’s instructions (cat# AC-06F1, Immunodiagnostic Systems, Gaithersburg, MD, USA).

*Breastmilk Calcium Assay*: Breastmilk collection was adapted from previously described methods [20]. Briefly, nursing dams were separated from their pups in the same cage for 2 to 4 h using a wire rack prior to breastmilk collection. Female mice were anesthetized using 2.5% isoflurane. A total of 1 U of oxytocin (cat# O4375, Sigma-Aldrich) in PBS was given intraperitoneally to induce letdown. Breastmilk was produced through gentle massaging of the teat and collected by a pipette. Approximately 100 μL of breastmilk was collected from each female. The calcium assay was performed according to the manufacturer’s instructions (cat# MAK022, Sigma-Aldrich).

*Quantitative PCR (qPCR):* Bone marrow cells (BMCs) were collected by cutting the distal end of the femur or tibia and spinning out the bone marrow at 21× *g* for 30 s. RBCs were lysed using BD Pharm Lyse™ (cat# 555899, BD Biosciences). RNA was extracted using Agilent Total RNA Isolation Mini Kit (cat#5185-5999, Agilent Technologies, Santa Clara, CA, USA). Gene expression changes were measured by qPCR using KAPA SYBR FAST One-Step Universal (cat#07959613001, Roche Diagnostics). Primers used are shown in Table 2.

*MicroCT data collection and analysis:* Femurs were scanned in μCT35 (Scanco Medical USA, Southeastern, PA, USA) at 55 kV, 72 μA, 4 W, at a resolution of 10 μm. Gauss sigma of 0.8, Gauss support of 1, lower threshold of 220 permilles for cancellous bone and 240 permilles for cortical bone, and upper threshold of 1000 permilles were used for all analyses. These thresholds correspond to 468.8, 592.2, and 3000 mg hydroxyapatite (HA)/cm^3^. For trabecular analysis, the start of the distal metaphyseal region of the femur was identified and a 1.5 mm region (150 slices) toward the proximal end was selected for evaluation. For cortical analysis, the midpoint of the femur was identified, and a 1.0 mm (100 slices) region was selected for analysis. Bone mineral density was obtained by quantitative μCT using phantoms for calibration [21].

*Statistical analysis:* All statistical significance tests were calculated using Prism software v. 9.5.0 (GraphPad Inc., La Jolla, CA, USA). We used non-parametric unpaired tests (e.g., two-tailed Mann–Whitney U and ANOVA) to avoid the assumption of normal distribution in our data.

## 3. Results

**IL-7 and IL-15 increase postpartum:** To characterize the state of the immune system, we determined the cytokines expressed by hematopoietically derived cells (CD45^+^) in the bone marrow. Cytokine gene expression was analyzed by qRT-qPCR. We found that IL-6, IL-7, IL-15, IL-17A, and TNFα have upregulated expression in lactating dams compared to age-matched nulligravida (NG) females (Figure 1A). All cytokines were expressed by CD45^+^ cells, except for TNFα, which was expressed in both subsets. Next, we investigated the source of IL-7. We harvested bone marrow cells (BMCs) from IL-7^eCFP^ reporter mice at d12 postpartum (PP), and the cyan fluorescent protein (CFP) expression was analyzed by flow cytometry. We observed a 6.7-fold increase in CFP^+^ DCs (Figure 1B) in PP females compared to NG females. In addition, we used immunofluorescence to assess the expression of IL-7 and IL-15 by CD11c^+^ cells in bone sections from virgin and PP mice. An increase in both CD11c^+^ cells and CD11c^+^ IL-7^+^ IL-15^+^ triple-positive cells was observed PP (Figure 1C). In summary, the data in Figure 1 shows increased levels of BMDCs that express IL-7 and IL-15 PP, consistent with our OVX results [15].

**Reduced apoptosis of IL-7^+^ DCs postpartum**: We previously showed that E_2_ regulates the levels of IL7^+^ DCs post-OVX by inducing apoptosis via Fas ligand (FasL). Here, we determined whether E_2_ also induces FasL postpartum. The total RNA was harvested from BMCs, and FasL expression was measured by qRT-qPCR. There was a 6.8-fold reduction in FasL expression in PP females (Figure 2A). E_2_ replacement given shortly after birth (between d0 and d2 postpartum) restored FasL expression back to levels found in NP females at d12. TUNEL staining showed a 10-fold decreased in apoptosis in CD11c^+^ DCs in the bone marrow of PP mice. With E_2_ replacement, the number of apoptotic cells returned to NG levels (Figure 2B). Taken together, these data demonstrate that induction of Fas ligand (FasL) expression by E_2_ is a conserved mechanism between postpartum and post-OVX [15]. 

**Differential regulation of dendritic cell subset by estrogen**: We also examined the effect of E_2_ loss on DC subsets to gain insights into DC dynamics during pregnancy and postpartum. Freshly isolated splenic DCs were cultured in the presence of 10 nM 17β-estradiol, or vehicle, and the cells were characterized via flow cytometry. The results showed that E_2_ favors conventional type 1 dendritic cells (cDC1s) and reduces the number of conventional type 2 dendritic cells (cDC2s) (Figure 2C). To determine whether the change in the levels of the two DC populations is due to apoptosis, we stained for annexin V. Our results show that E_2_ reduces the apoptosis of cDC1s but increases the apoptosis of cDC2s (Figure 2D). 

**TNFα and IL-17A expression correlates with bone resorption:** Our results in Figure 1 and Figure 2 show that E_2_ loss leads to increased levels of IL-7^+^DCs that also produce IL-15. To determine whether T_M_ activation by IL-15 and the subsequent production of TNFα and IL-17A contribute to bone resorption, we analyzed femurs from IL-15RA^ΔT^ mice d12 postpartum. Analysis of μCT showed that while both strains gained the same amount of bone during gestation (G), IL-15RA^ΔT^ mice had approximately 20% less trabecular bone resorption during lactation compared to the WT (Figure 3A). However, there was no difference in the cortical bone between the two strains (Figure 3B).

To validate the effect of IL-7 and IL-15 on activating T_M_, we purified bone marrow T-cells and cultured them for 48 h in the absence of CD3 and CD28 stimulation, with or without IL-7, IL-15, or both. IL-7 and IL-15 increased Ki-67 staining in T_M_, indicating increased proliferation in the presence of IL-7 and IL-15 (Figure 4A). Importantly, IL-7 and IL-15 lead to the expression of TNFα and IL-17A in the presence of both cytokines, which was not observed when T_M_ were incubated with either cytokine alone (Figure 4B). This antigen-independent, cytokine-dependent activation is a unique finding of our study.

To assess whether the low levels of TNFα and IL-17A produced by T_M_ postpartum promoted bone resorption, we examined the T-cell populations in the bone marrow of both the WT and IL-15RA^ΔT^ mice during PP using age-matched NG females of the same strain as controls. We observed that the percentage of T_M_ and the expression of TNFα and IL-17A were highly variable (Figure 4C) between the PP and control groups and the difference was not statistically significant. 

No differences were observed in litter sizes between WT and IL-15RA^ΔT^ mice. Both strains had litters of 5–6 pups per female. The growth of the pups is not linear—it occurs in spurts. Accordingly, the amount of milk depends on the demand. We hypothesized that TNFα^+^ or IL17A^+^ T_M_ may be dependent on the calcium demand. This would contribute to inter-individual sampling variability. To determine whether activated T_M_ contribute to bone resorption, we correlated serum CTX levels to the percentage of T_M_ that were positive for TNFα or IL-17A in the same animal. We observed a strong correlation between increased cytokine production and bone resorption in PP. In WT mice, TNFα and IL-17A explained 87.0% and 91.3% of the variance, respectively, with statistical significance (*p* < 0.01) (Figure 4D). In contrast, in IL-15RA^ΔT^ mice, these cytokines accounted for 22.2% and 16.2% of the variance, respectively, but this was not statistically significant (*p* > 0.1). No correlation was found in the NG controls for both strains (Figure 4D). Finally, the calcium concentration in the breas_tm_ilk of the IL-15RA^ΔT^ mice on day 12 postpartum relative to the WT mice was reduced by 28.6% (±2.8%) (Figure 4E). Together, our results indicate that T_M_ promote bone resorption postpartum, and in the absence of T_M_-mediated TNFα and IL-17, reduces calcium content in breastmilk.

## 4. Discussion

Previous studies have shown that calcium homeostasis during lactation is maintained by crosstalk between the mammary gland, the pituitary gland, and bone [22,23,24,25]. The infant triggers an oxytocin burst that leads to the release of PTHrP via prolactin. PTHrP activates bone resorption and the release of calcium into circulation via osteocytic and osteoclastic osteolysis [9]. The role of the hypoestrogenic state during lactation has not been elucidated. Here, we examined the role of E_2_ loss the on the memory T-cell (T_M_)-mediated production of TNFα and IL-17A in contributing to bone resorption and the calcium content of breastmilk. Consistent with our previous studies using ovariectomized mice, we observed that the CD45^+^ hematopoietic cell lineage is responsible for cytokine production in the bone marrow postpartum (Figure 1A). IL-7^eCFP^ reporter mice showed significantly increased CFP-positive cells in the CD45^+^ CD11b^+^ CD11c^+^ population in lactating dams (Figure 1B). We verified that the increased number of DCs was due to their increased lifespan via FasL expression in the bone marrow postpartum compared to nulliparous controls (Figure 2A). Reduced FasL expression resulted in decreased apoptosis in DCs (Figure 2B). E_2_ replacement treatment showed increased FasL expression as well as TUNEL^+^ cells, demonstrating that E_2_ could control DC levels (Figure 2B). 

The role of estrogen in the regulation of DC subsets by controlling their lifespan has not been previously documented. Our findings are consistent with the functional properties of cDC1s, whose main functional role is the cross-presentation of antigens that can also induce tolerance [26,27,28,29]. During pregnancy, a hyperestrogenic state, cDC1s would be favored as the predominant population to cross-present circulating fetal and placental antigens. In contrast, cDC2s are favored in the low-E_2_ state (i.e., postpartum and postmenopause) to promote immune-surveillance against pathogens and to tip the balance toward a proinflammatory environment [30].

To measure the effect of T_M_ on bone mass during lactation, we analyzed the femurs of WT and IL-15RA^ΔT^ mice harvested during gestation (G), during d12 postpartum (PP), and from age-matched NG littermate controls. We observed an increase in trabecular bone mass measured by BV/TV and BMD in both WT and IL-15RA^ΔT^ mice during gestation, consistent with previous findings [5,7]. During lactation, both the BV/TV and BMD in WT mice returned to NG levels. In IL-15RA^ΔT^ mice, PP mice had a higher BV/TV and BMD compared to the WTs (Figure 3A). No cortical bone accrual was seen between the G and NG mice for both strains. Cortical bone resorption is the same during lactation between the two strains, both decreasing by about 18% and 10% for Ct.Th and TMD compared to NG, respectively (Figure 3B). Data in Figure 3 indicates that IL-15 signaling in T_M_ contributes to trabecular but not cortical bone resorption during lactation. This is likely because trabecular resorption is predominantly osteoclastic while cortical resorption is osteocytic, which is primarily driven by PTHrP during lactation [8,9,10,11,12].

IL-7 and IL-15 together increased T-cell proliferation and cytokine production that is antigen-independent but cytokine-dependent (Figure 4A,B). While in culture and post-OVX these cytokines promoted TNFα and IL-17A expression in T_M_, the level of these cytokines was highly variable postpartum (Figure 4C). However, the level of TNFα and IL-17A expression correlated with bone resorption in WT mice but not in IL15RA^ΔT^ mice (Figure 4D) that do not express TNFα and IL-17A by T_M_. Further, a lower calcium concentration was found in the breastmilk of the IL15RA^ΔT^ mice (Figure 4E). Together, our data support the notion that T_M_-derived TNFα and IL-17A increases bone resorption to release calcium that contributes to the calcium in milk.

Typically, inflammation is considered to be a response to infection and/or pathogens. Our results indicate that low-grade inflammation plays a critical physiological role in releasing calcium from bone for lactation. The T_M_-mediated pathway appears to be activated during a hypoestrogenic state, both postpartum as well as postmenopause. Breastfeeding triggers the release of PTHrP from the mammary glands, which stimulates bone resorption to meet the demand for calcium. This fluctuation in serum calcium levels could result in intermittent increases in PTH, which has been shown to induce regulatory T-cells (T_REG_s) [31,32]. Thus, a potential mechanism of limiting T_M_ could be due to the induction of T_REG_s by iPTH. We attempted to test the suppression of TNFα^+^ and IL-17A^+^ T_M_ by administering oxytocin into OVX mice. However, the dosage of oxytocin as well as the time needed to observe phenotypic changes is not well established. These studies are underway and require further optimization.

A limitation of our study is the use of a single timepoint for data collection. We selected day 12 of the 21-day nursing period for mice because it coincided with the peak of milk production and the beginning of the pups’ transition to solid food [33,34]. In humans, breastfeeding inhibits the start of estrus cycling. This negative feedback is absent in mice, which can conceive while still nursing. To avoid potential confounding effects from the reactivation of ovarian function, we removed the male mice from the environment. However, differences in estrogen (E_2_) production between human and mouse ovaries may still pose a confounding factor in our results.

In summary, our study highlights the physiological role of immune system activation during the postpartum period, particularly through the regulation of calcium homeostasis by memory T-cells in conjunction with the oxytocin-PTHrP pathway. Our findings reveal for the first time the critical involvement of T-cells in reproductive processes in mammals. We propose that the immune-mediated regulation of bone resorption, triggered by E_2_ loss, likely evolved to ensure an adequate calcium supply in breastmilk. This mechanism ceases once lactation ceases and E_2_ levels normalize. The PTHrP produced by mammary glands releases Ca^++^ from the bone, over and above that released by low-grade inflammation, based on demand by feeding. However, the activation of this pathway occurs due to E_2_ loss postmenopause, but without attenuation, this leads to pathological osteoporosis. Understanding the immune system’s function in physiological contexts—beyond its role in controlling infections or tumors—provides a new insights into the role of the immune system.

## 5. Conclusions

Summarizing, we show that estrogen loss postpartum leads to expression of TNFα and IL-17A by memory T-cells via regulation of the lifespan of DC that express IL-7 and IL-15. Ablation of IL-15 receptor in T-cells, and consequently lack of TNFα and IL-17A reveals that these cytokines contribute 25 to 30% of calcium release for lactation. Further, estrogen loss is constant for the duration the ovaries are inactive postpartum leading to persistent low-grade expression of TNFα and IL-17A. Intermittent suckling by the pups activates oxytocin for milk letdown that triggers PTHrP to release calcium from production of milk for the next cycle of feeding. As pups feed every 2.5 to 3 hours [35], the PTHrP promotes fast cycling bone resorption than the constant low-grade inflammation.

## Figures and Tables

**Figure 1 nutrients-16-03289-f001:**
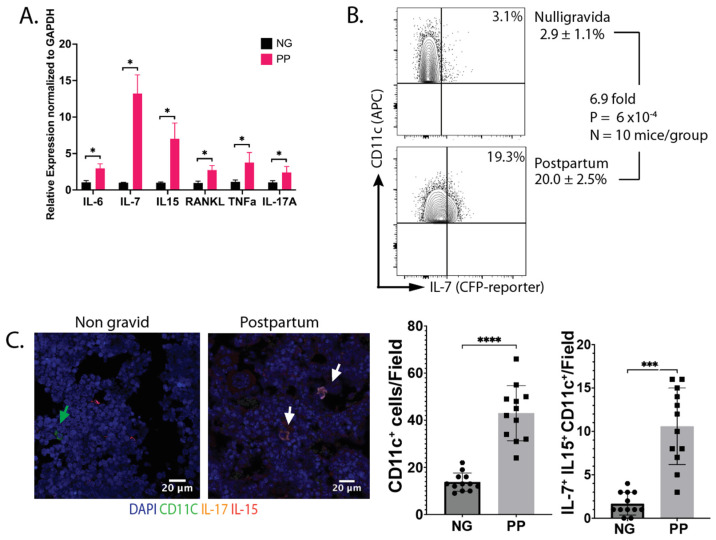
**The number of IL-7 and IL-15 expressing CD11c^+^ cells increases in the bone marrow postpartum.** (**A**) Total RNA was isolated from bone marrow cells (BMCs) from nulligravida (NG) and d12 postpartum (PP) C57BL6/J mice and analyzed by RT-qPCR. There is an increased expression in all cytokines in the PP mice compared to NG. Data are presented as relative expression normalized to GAPDH. Data were collected from 5 mice/group with 3 replicates/mouse. (**B**) BMCs from NG and PP IL-7^eCFP^ reporter mice were analyzed via flow cytometry, gated on CD45^+^, CD3^−^, CD11b^+^, CD11c^+^, and CFP^+^ cells. PP mice had a 6.7-fold increase in CFP^+^ cells compared to NG. (**C**) Both CD11c^+^ cells (green arrow) and CD11c^+^ IL-7^+^ IL-15^+^ triple-positive cells (white arrow) are increased in PP females compared to NG. Data are shown as the mean ± SEM. *p* values were calculated using unpaired two-tailed Mann–Whitney U test (* *p* < 0.1, *** *p* < 0.001, **** *p* < 0.0001).

**Figure 2 nutrients-16-03289-f002:**
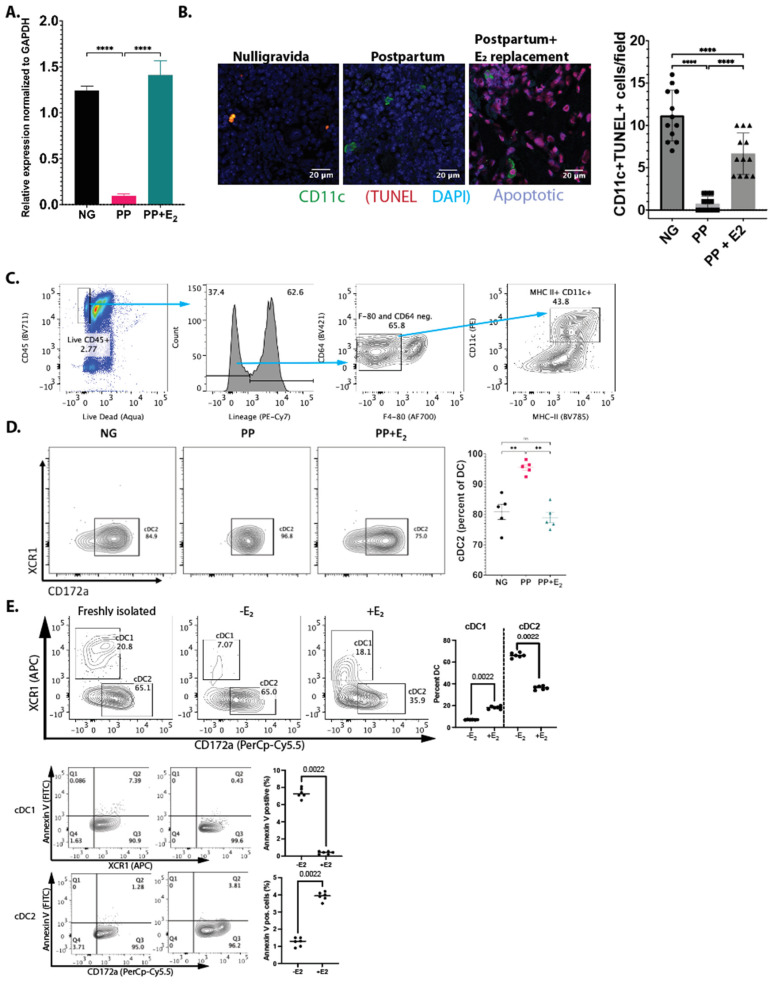
**The hypoestrogenic state postpartum regulate dendritic cell subsets.** (**A**) Total RNA was isolated from BMCs of NG, PP, and PP+E_2_ mice and FasL expression was quantified by RT-qPCR relative to GAPDH. FasL expression decreased in the BMCs in PP mice and returned to NG level with estrogen replacement using 0.05 mg 17β-estradiol pellets. Data were collected from 5 mice/group with 3 replicates/mouse. (**B**) Apoptosis in BMCs was visualized using TUNEL staining. The number of apoptotic foci (purple nuclei) in CD11c^+^ cells (green) decreased significantly in PP mice. Estrogen replacement increased the numbers of apoptotic foci, but they are still fewer than NG. (**C**) Splenic DCs were cultured for 4 h with or without 10 nM 17β-estradiol and populations were analyzed by flow cytometry. DCs were gated on CD45^+^ lineage^−^ (CD3^−^ CD19^−^ B220^−^ NK1.1^−^) F4/80^−^ CD64^−^ MHCII^+^ CD11c^+^ cells. cDC1s and cDC2s were further gated on XCR1^+^ and CD172a^+^ cells, respectively. The addition of E_2_ altered the proportion of cDC1s and cDC2s in favor of more cDC1s. (**D**) BMC were isolated from NG, PP, and PP+E_2_ mice and analyzed by flow cytometry. Only cDC2s were found in the bone marrow, which increased during PP and is reduced with E_2_ replacement (**E**) Annexin V staining revealed that E_2_ decreased apoptosis in cDC1s while simultaneously increasing apoptosis in cDC2s. Data are shown as the mean ± SEM. *p* values were calculated using unpaired two-tailed Mann–Whitney U test (ns = not significant, ** *p* < 0.01, **** *p* < 0.0001).

**Figure 3 nutrients-16-03289-f003:**
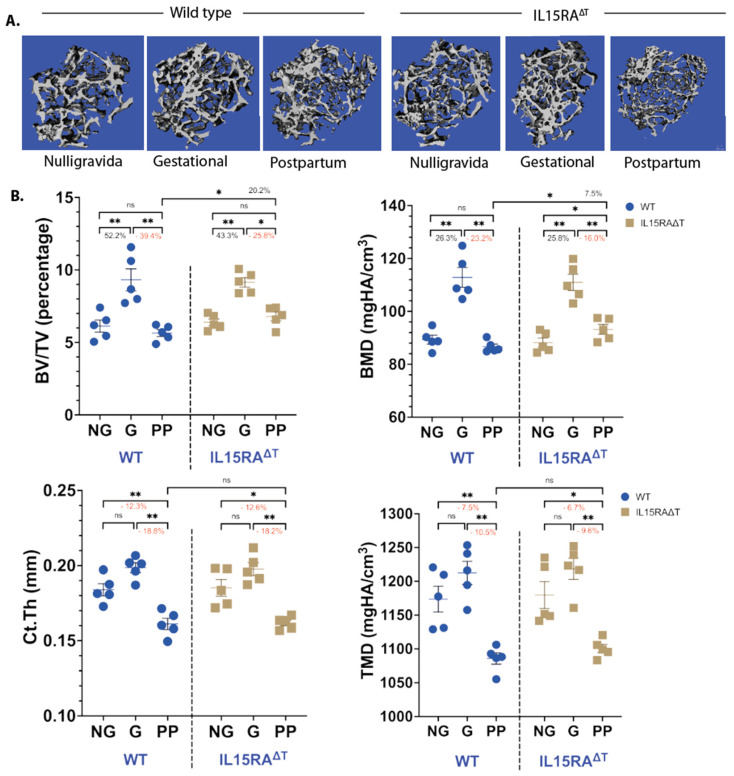
Specific ablation of IL-15 receptor α chain in T-cells reduces trabecular bone resorption but does not affect cortical bone postpartum. Femurs from nulligravida (NG), late gestational (G), and d12 postpartum (PP) mice were harvested and analyzed by microCT. (**A**) In WT mice, pregnancy results in increased bone volume-to-total volume ratio (BV/TV) and bone mineral density (BMD), which return to NG levels during lactation. During lactation, IL15RA^ΔT^ mice have reduced trabecular bone resorption compared to WT mice. (**B**) No difference is observed in both cortical thickness (Ct.Th) and tissue mineral density (TMD) between G and NG states in both WT and IL15RA^ΔT^ mice. Lactation results in similar decreases in Ct.Th and TMD in both strains. Data are shown as the mean ± SEM. *p* values were calculated using unpaired two-tailed Mann–Whitney U test (ns = not significant, * *p* < 0.1, ** *p* < 0.01).

**Figure 4 nutrients-16-03289-f004:**
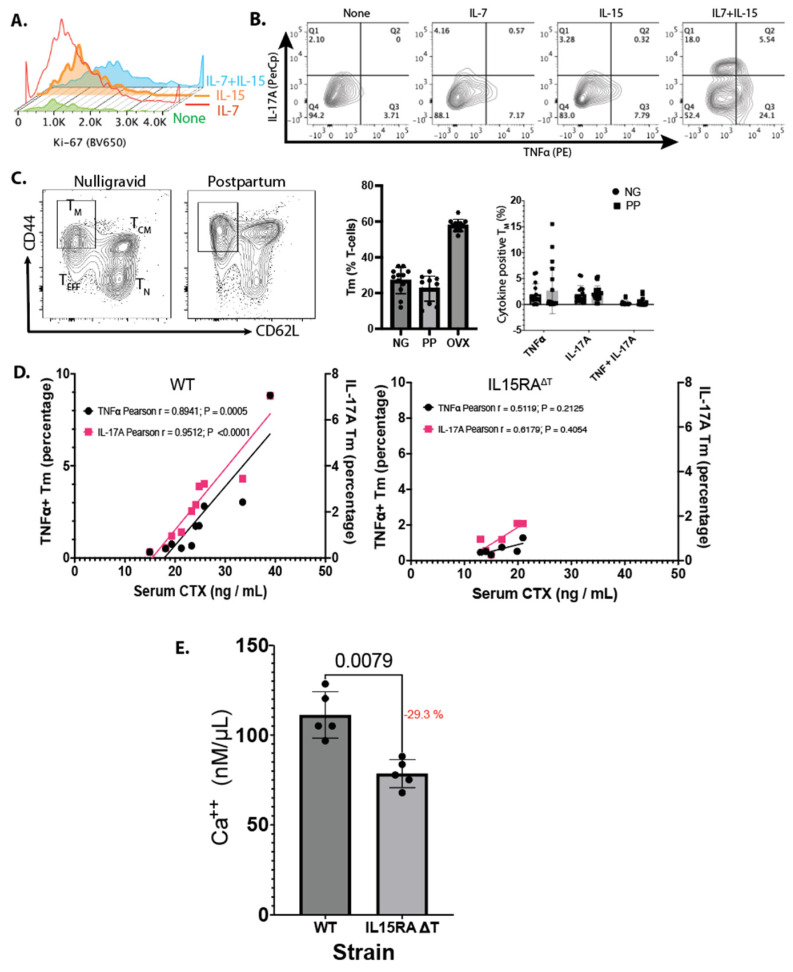
**T_M_ activation contributes to calcium release but the expression of TNFα and IL-17A is attenuate PP relative to OVX.** (**A**) T-cells were purified from BMCs and cultured in complete T-cell growth media for 48 h with or without IL-7, IL-15, or both. IL-7 alone increased Ki-67 staining in T_M_, while the presence of both significantly increased proliferation in the presence of IL-7 and IL-15. (**B**) IL-7 and IL-15 together led to increased expression of TNFα and IL-17A. (**C**) BMCs from WT and IL15RA^ΔT^ mice were harvested at 12d postpartum (PP) and analyzed via flow cytometry. Results from ovariectomized females are also shown for comparison. Age-matched nulliparous littermates were used as controls. (**D**) In WT mice, serum CTX levels showed a strong and statistically significant (*p* < 0.001) correlation with the percentage of T_M_ that expressed TNFα (87.0%) or IL-17A (91.38%) in the same animal. In IL-15RA^ΔT^ mice, these cytokines only explained 22.2% and 16.2% of the variance, respectively, with no statistical significance (*p* > 0.1). (**E**) Calcium concentration in the breastmilk of WT and IL-15RA^ΔT^ mice on day 12 postpartum showed that IL-15RA^ΔT^ mice had 28.6% (±2.8%) lower calcium concentration. Data are shown as the mean ± SEM. *p* values were calculated using unpaired two-tailed Mann–Whitney U test.

**Table 1 nutrients-16-03289-t001:** Antibodies used for flow cytometry.

Target	Clone	Fluorochrome
CD45	30-F11	FITC
CD3	17A2	PE-Cy7
CD19	1D3	PE-Cy7
B220	RA3-6B2	PE-Cy7
NK1.1	S17016D	PE-Cy7
CD4	RM4-5	PerCP-Cy5.5
CD4	RM4-5	ACP-CY7
CD8	53-6.7	BV510
CD8	53-6.7	ACP-CY7
CD44	IM7	AF700
CD62L	MEL+14	PacBlue
CD64	X54-5/7.1	BV605
CD69	H1.2F3	PE-Cy5
CD103	2E7	APC
F4/80	BM8	AF700
MHC II	M5/114.15.2	BV758
CD11c	N418	PE
XCR1	ZET	BV650
CD172a	P84	PerCP-Cy5.5
TNFα	MP6-XT22	PE
IL-17A	TC11-18H10	BV7585
Annexin V	n/a	AF488

**Table 2 nutrients-16-03289-t002:** PCR primers used in this study.

Primer	Sequence
GAPDH Fwd	CTGGAGAAACCTGCCAAGTA
GAPDH Rev	TGTTGCTGTAGCCGTATTCA
IL-6 Fwd	CTCTGGGAAATCGTGGAAAT
IL-6 Rev	CCAGTTTGGTAGCATCCATC
IL-7 Fwd	ACCATGTTCCATGTTTCTTTTAGAT
IL-7 Rev	TGCGAGCAGCACGATTTAGA
IL-15 Fwd	CGCCCAAAAGACTTGCAGTG
IL-15 Rev	GGTGGATTCTCTCTGAGCTGT
RANKL Fwd	CCCATCGGGTTCCCATAAAGT
RANKL Rev	CCCGATGTTT CATGATGCCG
TNFα Fwd	GTAGCCCACGTCGTAGCAAA
TNFα Rev	ACAAGGTACAACCCATCGGC
IL-17A Fwd	ATCCCTCAAAGCTCAGCGTGTC
IL-17A Rev	GGGTCTTCATTGCGGTGGAGAG
FasL Fwd	TTGGGCTCCTCCAGGGTCAGT
FasL Rev	CTGGGGTTGGCTCACGGAGT

## Data Availability

All data and detailed protocols are freely available upon email request to the corresponding author because there are no public repositories for the data type in this study.

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
