# Peer review of "Memory T-Cells Contribute to Calcium Release from Bones during Lactation in Mice"

_nutrients, 2024, doi:10.3390/nu16193289_

Round 1

Reviewer 1 Report

Comments and Suggestions for Authors

In the manuscript “Memory T cells contribute to calcium release from bones during lactation, Wu et al. examine the potential role for memory T cells and their derived cytokines in lactation-associated bone turnover. This hypothesis is a follow up to their work showing that estrogen deficiency rendered by ovariectomy promotes memory T cell production of inflammatory cytokines, which, in turn, is associated with increased bone turnover. Exploring the relationship between the immune cells and bone turnover during lactation is interesting and there are limited studies in the current literature. However, the experiments presented in this submitted manuscript are largely correlative, the characterization of in vivo experiments is incomplete and the conclusions drawn by the authors are overstated. 

Major weaknesses:

1.        The overall hypothesis is that during the hypoestrogenic state of lactation, proliferation of IL7+DC produce IL-7 and IL-15, which together, promote memory T cell proliferation and production of TNFaand IL-17, that, in turn, activates bone resorption. While the data presented show changes in levels of cytokines or distribution of immune cell populations in lactating animals, these data are observational and associations but not a direct test of the hypothesis. Furthermore, the in vitro data is not validated in the mouse models since the use of BM-derived cells should complement in vivo experiments – not substitute for them. 

2.        Experimental design is lacking detail. For the lactation studies, there is no mention of litter sizes and whether the litters were normalized across strains and studies. The variability in the production of key cytokines TNFa and IL-17 as well as milk calcium levels may be directly related to variable litter sizes. For the estrogen replacement studies, what was the dose of estrogen pellet used? What was the serum estrogen level? Experiments are not reproducible based on current descriptions. 

3.        A glaring deficiency is the lack of any measurement of bone mass changes.  CTX is a marker of bone resorption but not necessarily a substitute for directly measuring bone loss by PIXImus/DXA or QCT.  Also, it is traditional to perform static and/or dynamic histomorphometry to assess bone turnover and validate bone turnover markers. 

4.        If the central hypothesis is correct then Estrogen replacement should decrease IL7+IL15+ DC’s, TNFalpha and IL17 levels, CTX levels and mitigate bone loss.  These experiments should be done.  The IL-7 indicator mice would be useful for these experiments.

5.        A more thorough characterization of bone and mineral metabolism might explain why milk calcium levels decrease in the IL15 cKO mice, but CTX does not.  What happens to bone mass, circulating PTH or PTHrP levels ,or bone formation rates in these mice?

6.        The quality of the Figures is low and they are missing many labels. Figure 1C is missing labels for condition and representative images do not agree with the concomitant quantification. Figure 2B is missing labels—what are the conditions and stains? Figure 1A is missing statistics. In Figure 2, the text states N=15 mice/group but none of the graphs have 15 individuals represented. 

Minor weaknesses:

1.        The second introductory paragraph contains information about the physiology of lactation-associated bone turnover that is inaccurate (page 1, line 37-42).

2.        For the experiments in Figure 3, from which animals were the memory T cells derived? Nulliparous or lactating? Is there any priming of memory T cells from pregnancy or lactation that may play a role in its response to the hypoestrogenic state?

3.        The correlation between serum CTx and TNFa/IL-17 T memory cell study does not provide functional evidence that these cytokines are responsible for increased bone turnover in lactation. Furthermore, in the wildtype animals, there are 1-2 outliers that may be skewing the relationship. It is unclear to me how to interpret the correlation in the IL-15RA animals. In this group’s previous report (Ref 15), T cells deficient in IL-15RA do not express TNFa or IL-17. Here, there are no measurements of TNFa or IL-17 reported for the IL-15RA group. If there are TNFa or IL-17 positive T memory cells which are being corelated to CTX (Figure 3D), then this may represent an mechanism independent of IL-15/IL-7 or incomplete Cre recombination. But if IL15-RA T memory cells do not express TNFa or IL-17 (stated on page 9, line 332), then no correlation can be determined with CTX. This study also needs nulliparous control groups. 

4.        The Discussion tends to overstate the meaning of their results and there are statements that lack evidence. On page 9, line 338-340, “the Tm mediated pathway appears to be activated when PTHrP mediated bone resorption is insufficient to meet demand for calcium.” None of the experiments involved PTHrP or calcium insufficiency (e.g, low calcium diet). On page 9, line 342 “PTHrP reduces the activation and production of TNFa and IL-17A by Tm.” Again, none of the experiments presented involve PTHrP. 

5.        There are many grammatical and typographical errors throughout the text. 

Comments on the Quality of English Language

There are multiple grammatical errors and the manuscript would require editing for grammar.

Author Response

In the manuscript “Memory T cells contribute to calcium release from bones during lactation, Wu et al. examine the potential role for memory T cells and their derived cytokines in lactation-associated bone turnover. This hypothesis is a follow up to their work showing that estrogen deficiency rendered by ovariectomy promotes memory T cell production of inflammatory cytokines, which, in turn, is associated with increased bone turnover. Exploring the relationship between the immune cells and bone turnover during lactation is interesting and there are limited studies in the current literature. However, the experiments presented in this submitted manuscript are largely correlative, the characterization of in vivo experiments is incomplete and the conclusions drawn by the authors are overstated.

We thank the reviewer for the thorough reading of the manuscript and for the critiques. We appreciate that these critiques have greatly improved the strength of our manuscript.

Major weaknesses:

  1. The overall hypothesis is that during the hypoestrogenic state of lactation, proliferation of IL7+DC produce IL-7 and IL-15, which together, promote memory T cell proliferation and production of TNFα and IL-17, that, in turn, activates bone resorption. While the data presented show changes in levels of cytokines or distribution of immune cell populations in lactating animals, these data are observational and associations but not a direct test of the hypothesis. Furthermore, the in vitro data is not validated in the mouse models since the use of BM-derived cells should complement in vivo experiments – not substitute for them.

We include a graphical abstract to clarify our hypothesis: that memory T-cells (TM) are produce TNFα and IL-17A via IL-7 and IL-15 secreted by dendritic cells (DC). This pathway is activated by loss of estrogen (E2) postpartum and remains on until ovarian function restarts. Thus, we consider it to be a basal bone resorption postpartum. In a human, the ovaries are typically inactive while the female is breastfeeding. In contrast, in rodents the ovaries are re-activated postpartum by the presence of the male (within hours after delivery of the litter). Thus, we removed the male once female was pregnant. We focused on the consequences of E2 loss because this aspect has not been previously examined in detail. Our study shows that the pathway activated due to E2 loss in ovariectomized mice leading to osteoporosis (Ref. 15) is also used in the postpartum period to release calcium from the bone. The results thus establish the physiological role of TM in the postpartum period. The production of TNFα and IL-17A becomes pathological because it persists for decades postmenopause and not 6 to 12 months postpartum. All data shown is from in vivo lactating dams. Two exceptions are the response of DC to E2 in Fig. 2E. The regulation of DC is complex, and the levels of classical DC subsets varies by anatomical site. By isolating splenic DC, we tested for the effect of E2directly on both subsets, in the absence of confounding factors. The second exception is Fig. 4A where we test whether IL-7 and IL-15 together are sufficient to promote TNFα and IL-17A in TM.

  1. Experimental design is lacking detail. For the lactation studies, there is no mention of litter sizes and whether the litters were normalized across strains and studies. The variability in the production of key cytokines TNFα and IL-17 as well as milk calcium levels may be directly related to variable litter sizes. For the estrogen replacement studies, what was the dose of estrogen pellet used? What was the serum estrogen level? Experiments are not reproducible based on current descriptions.

All females we used in these experiments have a gravida of one. Control mice were nulliparous littermates. Litters were 5-6 pups per female. The dose of β-estradiol has been added (the pellets deliver 0.05 mg over 21-days). Additional details have been added to the methods section. We did not measure endogenous E2, as these values have been previously published.

  1. A glaring deficiency is the lack of any measurement of bone mass changes. CTX is a marker of bone resorption but not necessarily a substitute for directly measuring bone loss by PIXImus/DXA or QCT.  Also, it is traditional to perform static and/or dynamic histomorphometry to assess bone turnover and validate bone turnover markers.

We have included the μCT data as requested by the reviewer in the new Fig. 3. We did not include it because it provides no additional insights. Our key finding is that during lactation, the TM mediated inflammation provides a low basal level (~30%) release of calcium in breastmilk (Fig. 4E), whereas the remainder of calcium is provided by other mechanism, most likely via PTHrP. Further, using ovariectomy (OVX) to model E2 loss postmenopause we observe higher percent of TM in OVX than PP (Fig. 4C). This attenuation of the immune system during lactation suggests that there is mammary gland or parathyroid-dependent regulation of the TM. We will be investigating differences, targets and mechanism for the regulation of TM by E2 loss in PP and OVX in future experiments. 

  1. If the central hypothesis is correct then Estrogen replacement should decrease IL7+IL15+ DC’s, TNFalpha and IL17 levels, CTX levels and mitigate bone loss.  These experiments should be done. The IL-7 indicator mice would be useful for these experiments.

That experiment is shown in Fig. 2A. E2 replacement PP does increase FasL and apoptosis (TUNEL) of DCs bringing them back down. In comparison to OVX, where DC levels are restored to sham-surgery, here the levels do not come back to the full extent found in nulliparous females. This finding suggests that regulation of DC subsets may play an important role, as cDC2 produce more IL-15 that cDC1. This data will be presented elsewhere (manuscript in preparation).

  1. A more thorough characterization of bone and mineral metabolism might explain why milk calcium levels decrease in the IL15 cKO mice, but CTX does not.  What happens to bone mass, circulating PTH or PTHrP levels ,or bone formation rates in these mice?

Per the reviewer’s request, bone μCT parameters are now included in (new) Fig. 3. In line with our hypothesis, in the absence of TNFα and IL-17A by TM there is still significant bone loss in the cortical compartment but to lesser extent (~20%) in the trabecular compartment. This is consistent with prior experiments that PTHrP promotes bone remodeling in cortical bone through osteocytic osteolysis [PMID: 23109045]. The findings that PTHrP may promote bone resorption through osteocytes, while TM promote bone loss via osteoclasts may have implications for fracture risk in postmenopausal women and will be investigated in future studies.

  1. The quality of the Figures is low and they are missing many labels. Figure 1C is missing labels for condition and representative images do not agree with the concomitant quantification. Figure 2B is missing labels—what are the conditions and stains? Figure 1A is missing statistics. In Figure 2, the text states N=15 mice/group but none of the graphs have 15 individuals represented.

We apologize for the missing labels. We have corrected this in the revised version.

Minor weaknesses:

  1. The second introductory paragraph contains information about the physiology of lactation-associated bone turnover that is inaccurate (page 1, line 37-42).

Thank you for pointing out our error. We have corrected the paragraph

  1. For the experiments in Figure 3, from which animals were the memory T cells derived? Nulliparous or lactating? Is there any priming of memory T cells from pregnancy or lactation that may play a role in its response to the hypoestrogenic state?

The T-cells in (now) Fig. 4 A-B were isolated from nulliparous age-matched WT females. Fig 4C-D are in vivo from lactating or OVX mice.

  1. The correlation between serum CTx and TNFa/IL-17 T memory cell study does not provide functional evidence that these cytokines are responsible for increased bone turnover in lactation. Furthermore, in the wildtype animals, there are 1-2 outliers that may be skewing the relationship. It is unclear to me how to interpret the correlation in the IL-15RA animals. In this group’s previous report (Ref 15), T cells deficient in IL-15RA do not express TNFa or IL-17. Here, there are no measurements of TNFa or IL-17 reported for the IL-15RA group. If there are TNFa or IL-17 positive T memory cells which are being corelated to CTX (Figure 3D), then this may represent an mechanism independent of IL-15/IL-7 or incomplete Cre recombination. But if IL15-RA T memory cells do not express TNFa or IL-17 (stated on page 9, line 332), then no correlation can be determined with CTX. This study also needs nulliparous control groups.

Fig. 4D shows the correlation between TNFα and IL-17 with CTX from IL15RAΔT mice. It shows a very low background levels (near the lower limit of detection) of TNFα and IL-17A by TM. In contrast, in wild-type mice there was a strong correlation between TNFα, IL-17A and CTX. The reviewer’s point (here in Major #3 above) does bring up the consideration of time scales. Estrogen loss is constant for the duration the ovaries are inactive postpartum leading to persistent low-grade TNFα and IL-17A. Intermittent suckling by the pups activates oxytocin for milk letdown that triggers PTHrP to release calcium from production of milk for the next cycle of feeding. As pups feed every 2.5 to 3 hours [PMID: 32201708], the PTHrP promotes fast cycling bone resorption than the constant low-grade inflammation.

  1. The Discussion tends to overstate the meaning of their results and there are statements that lack evidence. On page 9, line 338-340, “the Tm mediated pathway appears to be activated when PTHrP mediated bone resorption is insufficient to meet demand for calcium.” None of the experiments involved PTHrP or calcium insufficiency (e.g, low calcium diet). On page 9, line 342 “PTHrP reduces the activation and production of TNFa and IL-17A by Tm.” Again, none of the experiments presented involve PTHrP. 

See Fig 4C and rebuttal to point 3 above.

  1. There are many grammatical and typographical errors throughout the text. 

We have proofread and tried to eliminate most typographical errors.

Reviewer 2 Report

Comments and Suggestions for Authors

The manuscript is interesting and also relevant. I tried to find the main question they want to answer but could not find it. The only sentence about that is: Our studies reveal a new mechanism for the release of calcium from bone during breast feeding. Because the text is not easy to follow I suggested a sort of graphical abstract to make all the pathways they found or suggest more clear. Thus the text is not easy to read. The experiments they did are of good quality, but the presented figures are sometimes very small. I think this finding can add a new inside in the mechanism of bone degradation during lactation and maybe also osteoporosis (but they did not look or speculate on that)

The manuscript has a lots of data but that makes it also hard to read. Some figures are small and not very consistent. In the text the mice mentioned differently. postpartum, lacting dams, nursing dams or virgin or nulliparous.  Is nice to use not always the same words in the text but makes it hard to read.

I suggest that a graph (or graphical abstract) with the whole pathway makes it much easier to read and follow the steps they are made.

What has to be improved:

- Some figures larger

- What is the main question that has to be addressed

- graph to make the pathways more clear

- Don't use different names for the 2 groups of mice used.

I suggested revisions because the experiments done are nice but the text needs revision.

Author Response

The manuscript is interesting and also relevant. I tried to find the main question they want to answer but could not find it. The only sentence about that is: Our studies reveal a new mechanism for the release of calcium from bone during breast feeding. Because the text is not easy to follow. I suggested a sort of graphical abstract to make all the pathways they found or suggest more clear. Thus the text is not easy to read. The experiments they did are of good quality, but the presented figures are sometimes very small. I think this finding can add a new inside in the mechanism of bone degradation during lactation and maybe also osteoporosis (but they did not look or speculate on that).  I suggest that a graph (or graphical abstract) with the whole pathway makes it much easier to read and follow the steps they are made.

We thank the reviewer for recognizing the relevance of our study. We have reworded the abstract and included a graphical abstract as recommended to clarify the central question.

The manuscript has a lots of data but that makes it also hard to read. Some figures are small and not very consistent. In the text the mice mentioned differently. postpartum, lacting dams, nursing dams or virgin or nulliparous.  Is nice to use not always the same words in the text but makes it hard to read.

We understand. We have tried to be more consistent in the revised version.

What has to be improved:
- Some figures larger
- What is the main question that has to be addressed
- graph to make the pathways more clear
- Don't use different names for the 2 groups of mice used.

I suggested revisions because the experiments done are nice but the text needs revision.

We have revised the manuscripts and used the reviewer’s guidance. We hope that the revised version is improved.

Reviewer 3 Report

Comments and Suggestions for Authors

The paper described how estrogen (E2) depletion postpartum affect the production of TNFalpha and IL-17A by memory T-cells and contribute to calcium release from bones during lactation. The relationship between E2 loss, T cells and calcium content of breastmilk is well described, while the connection with calcium release from bones is less clear. In the methods section there is a " MicroCT data collection and analysis" paragraph, but no results were reported. Without these results, changes in calcium levels could be ascribable to other phenomena (lower renal retention or intestinal absorption?).     Minor issues:   pag 1 line 36 "Recent studies " ref 6-8 are not so recent. Please add more recent references or modify the sentence.   page 2 line 67 typo error IL-7 instead of L-7

pag 4 Methods: primers description is missing

Author Response

The paper described how estrogen (E2) depletion postpartum affect the production of TNFalpha and IL-17A by memory T-cells and contribute to calcium release from bones during lactation. The relationship between E2 loss, T cells and calcium content of breastmilk is well described, while the connection with calcium release from bones is less clear. In the methods section there is a " MicroCT data collection and analysis" paragraph, but no results were reported. Without these results, changes in calcium levels could be ascribable to other phenomena (lower renal retention or intestinal absorption?).

We have included the microCT data in the revised version. 

Minor issues:   pag 1 line 36 "Recent studies " ref 6-8 are not so recent. Please add more recent references or modify the sentence. page 2 line 67 typo error IL-7 instead of L-7

We have revised the sentences.

Reviewer 4 Report

Comments and Suggestions for Authors

The problem describing in the manuscript is interesting, well written.

The study are focused on mice and in the title it should be added what was the experimental model.

Why the serum level was not measured? Three is no information about the hormonal status of the HPG axis.

After title the full stop should be taken out.

Conclusion - should be written clearly, also the side effects of estradiol supplementation should be included.

Author Response

The problem describing in the manuscript is interesting, well written.

We thank the reviewer for recognizing the interesting nature of the manuscript.

The study are focused on mice and in the title it should be added what was the experimental model. After title the full stop should be taken out.

We have revised the title as recommended by the reviewer.

Why the serum level was not measured? Three is no information about the hormonal status of the HPG axis.

Multiple studies have established that postpartum the ovaries are inactive in humans while the mother is breastfeeding. In rodents the ovaries stay inactive postpartum in the absence of a male. Here we focused the consequences of estrogen loss (due to the inactivity of the ovaries) because it has not been studied in detail. Our study shows that the pathway activated due to estrogen loss in ovariectomized leading to osteoporosis is also used in the postpartum period to release calcium from the bone. The results thus establish the physiological role of memory T-cells in the postpartum period. The production of TNFα and IL-17A becomes pathological because it persists for decades postmenopause (not months in postpartum) and possibly because it is not attenuated by the HPG axis (Fig. 4C)

Conclusion - should be written clearly, also the side effects of estradiol supplementation should be included.

There were no side effects of estrogen replacement noted. We expect that estrogen levels would be physiologically restored in these mice after we put a male in the cage. We have revised the manuscript to clarify our conclusions.

Round 2

Reviewer 3 Report

Comments and Suggestions for Authors

Authors reply to all the observations. I only suggest to modify Fig 1 in graph legend "triple cells" by making explicit CD11c+ IL-7+ Il-15+ to help readers.